# Analysis of the Probiotic Potential of *Lactiplantibacillus plantarum* LB1_P46 Isolated from the Mexican Fermented Pulque Beverage: A Functional and Genomic Analysis

**DOI:** 10.3390/microorganisms12081652

**Published:** 2024-08-12

**Authors:** Martha Giles-Gómez, Ximena Morales Huerta, Rodolfo Pastelin-Palacios, Constantino López-Macías, Mayrene Sarai Flores Montesinos, Fernando Astudillo-Melgar, Adelfo Escalante

**Affiliations:** 1Departamento de Biología, Facultad de Química, Universidad Nacional Autónoma de México, Ciudad de México 04510, Mexico; margiles@quimica.unam.mx (M.G.-G.); ximohu@gmail.com (X.M.H.);; 2Unidad de Investigación Médica en Inmunoquímica, Hospital de Especialidades del Centro Médico Nacional “Siglo XXI”, Instituto Mexicano del Seguro Social (IMSS), Cuauhtémoc, Ciudad de México 06720, Mexico; 3Departamento de Ingeniería Celular y Biocatálisis, Instituto de Biotecnología, Universidad Nacional Autónoma de México, Cuernavaca 62210, Morelos, Mexico; mayrene.flores@ibt.unam.mx (M.S.F.M.);

**Keywords:** *Lactiplantibacillus plantarum*, pulque, probiotic lactic acid bacteria, antimicrobial activity, anti-infective activity, bacteriocins, plantaricin EF, RiPPs

## Abstract

The traditional Mexican fermented beverage pulque has been considered a healthy product for treating gastrointestinal disorders. Lactic acid bacteria (LAB) have been identified as one of the most abundant microbial groups during pulque fermentation. As traditional pulque is consumed directly from the fermentation vessel, the naturally associated LABs are ingested, reaching the consumer’s small intestine alive, suggesting their potential probiotic capability. In this contribution, we assayed the probiotic potential of the strain of *Lactiplantibacillus plantarum* LB1_P46 isolated from pulque produced in Huitzilac, Morelos State, Mexico. The characterization included resistance to acid pH (3.5) and exposure to bile salts at 37 °C; the assay of the hemolytic activity and antibiotic resistance profiling; the functional traits of cholesterol reduction and β-galactosidase activity; and several cell surface properties, indicating that this LAB possesses probiotic properties comparable to other LAB. Additionally, this *L. plantarum* showed significance in in vitro antimicrobial activity against several Gram-negative and Gram-positive bacteria and in vivo preventive anti-infective capability against *Salmonella* in a BALB/c mouse model. Several functional traits and probiotic activities assayed were correlated with the corresponding enzymes encoded in the complete genome of the strain. The genome mining for bacteriocins led to the identification of several bacteriocins and a ribosomally synthesized and post-translationally modified peptide encoding for the plantaricin EF. Results indicated that *L. plantarum* LB1_P46 is a promising probiotic LAB for preparing functional non-dairy and dairy beverages.

## 1. Introduction

Pulque is a traditional Mexican alcoholic non-distilled fermented beverage produced from the fermentation of the fresh sap (aguamiel) extracted from several *Agave* (maguey) species, mainly *Agave americana*, *A. salmiana*, *A. mapisaga* and produced mainly in the Central Mexican Plateau. Pulque is a milky white, viscous, acidic, and slightly alcoholic (less than 6 °GL) beverage produced in Mexico since the pre-Hispanic period. It is the most studied traditional fermented beverage because of its historical, religious, economic, social, and medical relevance. Additionally, maguey cultivation for pulque production maintains a relevant traditional agroforestry significance [1,2,3].

Studies on the microbiology of pulque have focused on identifying those microorganisms responsible for the fermentation of the sap to yield pulque and the development of the final sensorial properties of the beverage. Pioneer studies on the microbiology of pulque reported *Lactobacillus*, *Leuconostoc*, *Zymomonas*, and *Saccharomyces* as the main microorganisms responsible for fermentation of the main sugars (sucrose, glucose, and fructose) present in aguamiel, resulting in the development of lactic acid, alcoholic fermentation, and the production of extracellular polysaccharides like dextran, responsible of the characteristic acidic, alcoholic, and viscous properties of pulque [4]. Further studies have led to the detection of a great diversity of lactic acid bacteria (LAB) in aguamiel and pulque [5,6]. Recent applications of metagenomics and amplicon sequencing (variable regions of the 16S rDNA gene and ITS regions) led to determining the complete microbial composition of aguamiel and pulque samples [7,8,9,10], proposing for the first time the dynamics of the microbial diversity during aguamiel fermentation to produce pulque, and proposing a microbial core responsible for the fermentation, highlighting the abundance of LAB during the fermentation process [7,8]. 

Traditional fermented foods and beverages have been considered a source of a great diversity of LAB, many identified as potential probiotic bacteria [11,12,13,14]. Several LAB strains of lactobacilli, *Lactococcus lactis*, *Leuconostoc* spp., *Pediococcus* spp., and *Streptococcus thermophilus*, have been included in the latest relation of Qualified Presumption of Safety (QPS) by the European Food Safety Authority (EFSA), providing a safety assessment approach for microorganisms intended for use in food or feed chains [15]. Additionally, the Food and Drug Administration (FDA) included in the Recently Published GRAS Notices and FDA Letters for food and beverages as regulated products to *Lactiplantibacillus plantarum*, *Lacticaseibacillus rhamnosus*, *S. thermophilus*, and *Lactobacillus acidophilus*, among other LAB [16].

Pulque has traditionally been considered a functional-like beverage due to the different benefits associated with its consumption [3,17,18]. Recent studies have focused on the study of potential probiotic benefits of several LAB isolated from aguamiel and pulque from diverse geographical origins, including the in vitro anti-infective capabilities of an isolate of *L. mesenteroides* against enterobacterial bacteria like *Escherichia coli*, *Salmonella enterica* subsp. *enterica* serovar Typhi (*Salmonella* Typhi), *Salmonella enterica* subsp. *enterica* serovar Typhimurium (*Salmonella* Typhimurium), and the Gram-positive pathogen *Listeria monocytogenes*; and in vivo anti-infective assays in a mouse model against *Salmonella* Typhimurium [19]. The administration of *L. sanfranciscensis* LBH1068 in a mouse model resulted in a significant reduction in weight loss, a decrease in gut permeability, and cytokine modulation resulted in improved mouse health [20]. Other studies include the in vitro activity against *Staphylococcus aureus* and *Helicobacter pylori* by *Lactobacillus* spp. and *Pediococcus* spp. [21] and the in vitro antimicrobial activity of *L. plantarum*, *Levilactobacillus brevis*, and *Lacticaseibacillus paracasei* against *Listeria monocytogenes* [22]. The consumption of aguamiel and pulque has been demonstrated as a source of vitamins and minerals [23,24,25]. Additionally, the metagenomic analysis of aguamiel and pulque also demonstrated the enrichment of the biosynthetic metabolic pathways for the synthesis of folates [8].

*Lactiplantibacillus plantarum*, previously designated as *Lactobacillus plantarum*, is a LAB isolated from diverse fermented foods, including fermented vegetables, meats, dairy products, and cereals. *L. plantarum* has relevant commercial importance as a starter culture for food fermentations and as a probiotic bacteria in humans and animals for its ability to survive gastric transit and colonize the intestinal tract of consumers [26,27]. *L. plantarum* produces several bacteriocins known as plantaricins, used as food preservatives and starters for dairy, meat, and fish fermentations, showing a diverse spectrum of antimicrobial activity against several Gram-positive and Gram-negative bacteria, including pathogenic bacteria [11,28]. Plantaricins include plantaricin A, produced by *L. plantarum* C11, showing narrow activity against other LAB such as *L. casei*, *L. sakei*, *L. viridescens*, and other *L. plantarum.* Plantaricin C has a broad spectrum against several Gram-positive bacteria. Other plantaricins, like the plantaricin C19 produced by *L. plantarum* isolated from traditional fermented foods, have broad inhibitory activity against pathogenic *Listeria monocytogenes.* The plantaricin 35d showed antimicrobial activity against *Staphylococcus aureus*, *L. monocytogenes*, *Aeromonas hydrophila*, and the plantaricin MG showed activity against *L. monocytogenes*, *S. aureus*, *Salmonella* Typhimurium, and *E. coli* [28]. Because of the relevance of this LAB, several *L. plantarum* strains have been isolated from traditional fermented sources from India, Indonesia, Turkey, and Mexico and assayed for probiotic capabilities [22,29,30,31].

In the present contribution, we report functional probiotic traits and the in vitro and in vivo antimicrobial and anti-infective activity of the *Lactiplantibacillus plantarum* strain LB1_P46. As we previously reported the sequence of the complete genome of two *L. plantarum* isolates from traditional Mexican fermented pulque [32], we correlated several assayed functional probiotic characteristics with the encoded proteins in the genome possibly involved with the antimicrobial and potential probiotic properties. A genome-mining analysis identified several genes for bacteriocins and ribosomally synthesized and post-translationally modified peptides (RiPPs).

## 2. Materials and Methods

### 2.1. Bacterial Strains and Cultivation Conditions

*Lactiplantibacillus plantarum* LB1_P46 was isolated from fermented pulque beverage produced in the municipality of Huitzilac (19°01′42′′ N 99°16′02′′ W), Morelos State, Mexico, as described previously. Its identity was determined by whole genome sequence comparison in the BLAST application at NCBI, assigning the identity of *L. plantarum* (query cover 100%, identity 99.80%). The assembled sequence of the isolate LB1_P46 was deposited in the GenBank database under BioProject PRJNA1021245, Biosample SAMN37547333 (LB1_P46) [32]. This LAB was cultivated routinary in MRS broth (DIFCO) and agar MRS (DIFCO) overnight at 30 °C or 37 °C, as required.

Bacterial strains used to assay the in vitro antimicrobial activity were provided by the culture collection of Facultad de Química-Universidad Nacional Autónoma de México. *Salmonella enterica* subsp. *enterica* serovar Typhimurium ATCC14028, *S. enterica* subsp. *enterica* serovar Typhi ATCC9992, *Pseudomonas aeruginosa* ATCC27853, and *Escherichia coli* 112 were grown in LB broth or agar (DIFCO). *Staphylococcus aureus* ATCC6538, *Listeria monocytogenes* CFQ-B-103, *Streptococcus pyogenes* CFQ-B-128, *Enterococcus faecalis* CFQ-B60, and *B. cereus* CFQ-B-230 were grown in BHI broth or agar (DIFCO). The streptomycin-resistant *Salmonella enterica* subsp. *enterica* serovar Typhimurium L1334 (str^r^) was used for in vivo anti-infective activity assays and grown in LB broth or plates (DIFCO) supplemented with 100 μg/mL streptomycin [19].

### 2.2. Safety Traits of L. plantarum

#### 2.2.1. Hemolytic Activity

As reported by [22], the hemolytic activity was determined by streaking a colony of *L. plantarum* and the probiotic control bacteria in Columbia agar medium supplemented with 5% defibrinated blood in quadruplicate and incubated at 37 °C for 24 h. Expected hemolytic activity was the appearance and color of a halo: transparent, β-hemolytic activity; green, α-hemolytic; no halo, no activity.

#### 2.2.2. Antibiotic Resistance

The assay for antibiotic resistance of the *L. plantarum* LB1_P46 was performed using the Antibiotic Susceptibility Testing Strips (ATB™ STREP 5 test strips, bioMérieux, Marcy-l’Étoile, France). The strain was grown in MRS broth as described until it reached a turbidity equivalent to 1 unit of the standard of McFarland. A quantity of 200 μL of the previous suspension was inoculated in the ATB S medium (bioMérieux) as described by the provider. A quantity of 135 μL of this solution was inoculated in each well of the ATB STREP 5 and was incubated at 37 °C for 24 h. The results were analyzed by registering the development (resistance to antibiotics) or absence (sensibility) of turbidity in the wells, determining if the analyzed strains were antibiotic sensitive, intermediate sensitive, or resistant. The Multiple Antibiotic Resistance (*MAR*) index (Equation (1)) was calculated as follows, according to [29]: (1)$$MAR\;index=\frac{Number\;of\;antibiotic\;that\;isolate\;is\;resistant\;}{Total\;numer\;of\;antibiotics\;tested}$$

### 2.3. Functional Traits

#### 2.3.1. Cholesterol Reduction

The cholesterol reduction assay was performed as reported by [22]. The *L. plantarum* strain was grown in MRS broth at 37 °C for 20 h. Meanwhile, the MRS–cholesterol broth was prepared by adding 0.5% bile salts (Sigma-Aldrich, Naucalpan de Juárez, Mexico) and 100 μg/mL of cholesterol (Sigma-Aldrich dissolved in ethanol (J. T. Baker, Xalostoc, Mexico). Analyzed strains were inoculated in the MRS–cholesterol broth with an inoculum of 1% *v*/*v* and incubated for 24 h at 37 °C. The control condition was MRS–cholesterol broth inoculated with 1% *v*/*v* of MRS broth. Grown cultures were centrifuged at 3000× *g* for 10 min, the supernatants were recovered, and the absorbance was determined at 570 nm. The percent of cholesterol reduction (*CR*) (Equation (2)) was calculated as follows: (2)$$CR\;(\%)=\left(\frac{A_{blank}-A_{LAB}}{A_{blank}}\right)\times 100$$
where *A_blank_* and *A_LAB_* are, respectively, the absorbance of the blank and the *LAB* supernatant cell suspension.

#### 2.3.2. β-Galactosidase Activity

The β-galactosidase activity was assayed as reported previously [22]. *L. plantarum* LB1_P46 was streaked in quadruplicate in lactose medium plates (0.5% lactose, 0.5% peptone, 0.3% meat extract, and 1.5% bacteriological agar) supplemented with 50 μg/mL of 5-bromo-4-chloro-3-idolyl-β-D-galactopyranoside (X-gal) (Sigma-Aldrich). Plates were incubated at 37 °C for 24–48 and analyzed by the presence of blue-stained colonies indicative of β-galactosidase activity.

### 2.4. Cell Surface Properties

#### 2.4.1. Hydrophobicity

The hydrophobicity (H) assay was determined by analyzing the Microbial Adhesion To Solvents (MATS) as determined previously [22] with slight modifications: Mild-exponential growing cultures in MRS broth at 37 °C of the strain LB1_P46 were centrifuged at 5000× *g* for 30 min, followed by two washes PBS (pH 7.2) and resuspended in this buffer at an OD_600 nm_ = 1.0. The resultant cellular suspension was separately mixed with chloroform and hexane 1:1 *v*/*v*) and vortexed for 30 s, then incubated for one hour at room temperature, and the DO_600 nm_ was measured. The *H*-value (Equation (3)) was calculated by the following: (3)$$H(\%)=\left(\frac{A_{0}-A}{A_{0}}\right)\times 100$$
where *A*_0_ and *A* are, respectively, the absorbance determined by DO_600 nm_ before and after treatment with organic solvents. 

#### 2.4.2. Auto-Aggregation

This assay was performed as previously reported [22]: Cells grown in MRS broth were harvested at the mild exponential phase and centrifuged as above. The resultant pellet was twice washed in PBS pH 7.2 and resuspended in PBS to an OD_600 nm_ = 0.5. This cellular suspension was incubated at 25 °C, and auto-aggregation was monitored by measuring the absorbance at 600nm at 0, 2, 4, 6, 20, and 24 h. Auto-aggregation (*A*) (Equation (4)) was calculated as follows: (4)$$A(\%)=\left(\frac{A_{0}-A}{A_{0}}\right)\times 100$$
where *A*_0_ and *A* indicate the absorbance before and after extraction with the organic solvents used.

#### 2.4.3. Coaggregation 

The coaggregation assay was assayed as reported previously [22]: Bacterial cell suspensions were prepared as previously for the auto-aggregation assay and mixed separately with an equal volume of the above-described *L. monocytogenes*, *Salmonella* Typhi and *E. coli* and incubated at 25 °C. Coaggregation was monitored by measuring the OD_600 nm_ at 0, 2, 4, 6, 20, and 24 h. Coaggregation (*C*) (Equation (5)) was calculated as follows:(5)$$C(\%)=\left[\frac{\left(A_{X}+A_{LAB}\right)-(A_{mix})}{\left(A_{X}+A_{LAB}\right)}\right]\times 100$$
where *A_X_* is the initial absorbance (DO_600 nm_) of *L. monocytogenes*, *Salmonella* Typhi, or *E. coli*; *A_LAB_* is the initial absorbance (DO_600 nm_) of the assayed *LAB* strain, and *A_mix_* is the absorbance (DO_600 nm_) of the mixture at different times.

All the assays described in Section 2.3 and Section 2.4 were performed in biological triplicates, and the commercial probiotic *Lactobacillus casei* Shirota, isolated from a commercial probiotic beverage, was used as a positive probiotic control.

### 2.5. In Vitro Assays 

#### 2.5.1. Bile Salt and Acid Resistance

Bile salt and acid resistance assays were assayed as reported previously by [19] with slight modifications. An overnight preculture of the strain LB1_P46 in 5 mL MRS broth was incubated for acid resistance at 37 °C. Resultant growth was adjusted to a final DO_600 nm_ = 0.2 in 0.5 mL to inoculate a new culture of 4.5 mL MRS acidified to pH 3.5 with 1N HCl (Baker) and incubated at 37 °C. Growth was monitored at 4 and 24 h by plating an aliquot of the culture in agar MRS plates incubated as above. For the bile salt resistance assay, an inoculum of 0.5 mL of the above culture of the acid resistance assay was used to inoculate 4.5 mL of MRS broth supplemented with 0.3% bile salts (oxgall, Oxoid, Basingstoke, UK), incubated at 37 °C for 4 and 24 h, and growth was monitored as above. Additionally, an assay that included acidified MRS (pH 3.5) supplemented with 0.3% bile salts was performed and monitored as above. All experiments were performed in biological duplicates, and the commercial probiotic *Lactobacillus casei* Shirota isolated from a commercial probiotic beverage was used as a positive control. 

#### 2.5.2. Antibacterial Assays against Pathogenic Bacteria

In vitro antibacterial activity of *L. plantarum* LB1_P46 isolate was assayed against pathogenic bacteria as reported by [19]: An aliquot of 0.1 mL of a cell suspension containing 9 log CFU/mL of an overnight culture of strains LB1_P46 and LB1_P47 grown in APT was dropped onto two fresh APT plates in quadruplicate for each plate for each assayed pathogenic bacterium and incubated overnight at 37 °C. The resultant spotted growth was overlaid with 5 mL of BHI or LB soft agar containing 0.5 mL of the overnight culture of assayed pathogenic bacteria adjusted to an optical density of OD_600 nm_ = 0.2. The plates were incubated overnight at 37 °C and evaluated for antibacterial activity by measuring each zone of inhibition with a millimeter ruler around the spotted growth of *L. plantarum.*

#### 2.5.3. In Vivo Preventive Anti-Infective Activity of *L. plantarum* LB1_P46 Isolate against *Salmonella* Typhimurium

The in vivo preventive anti-infective activity of *L. plantarum* LB1_P46 against the streptomycin-resistant (str^r^) *Salmonella* Typhimurium strain L1334 was assayed in male BALB/c mice with 8–9 weeks old in the vivarium of the Departamento de Medicina Experimental de la Facultad de Medicina, UNAM—Hospital General de México. The experiment was performed as described previously with several modifications [19]. Three groups were assayed: the PBS group without administration of LAB, the LB1_P46 group, and the *L. casei* Shirota group (probiotic bacteria group). All the groups comprised ten mice each. Strain LB1_P46 and *L. casei* Shirota were grown as described above, centrifuged (5000× *g*, for 5 min at 4 °C), and resuspended in 1 mL of sterile PBS buffer (0.8% NaCl, 0.121% K_2_HPO_4_, 0.034% KH_2_PO_4_, pH 7.4), and the cell density was adjusted to 9.3 log CFU/mL in 300 μL per dose. The experiment was carried out as follows in five stages: (1) adaptation of mice to the laboratory environment for four days, (2) daily administration of *L. plantarum* LB1_P46 or *L. casei* Shirota with an esophagogastric probe for seven days, (3) one resting day without bacterial administration, (4) infection at the next day with *Salmonella* Typhimurium adjusted to 7 log CFU/mL with an esophagogastric probe, and (5) after two days, mice were sacrificed, and the target blank organs (liver and spleen) were extracted and analyzed. Throughout the experiment, mice were provided with food and water ad libitum. The mice infection group was treated as described for experimental and probiotic groups, and PBS buffer was administrated before the infection with *Salmonella*. Organ extraction and processing were performed as previously [19], and cell suspensions were serially diluted in sterile saline and plated on agar LB supplemented with 100 mg/mL streptomycin (Sigma-Aldrich). The total log CFU/mL was determined by duplicate after incubation at 37 °C for 24 h. 

### 2.6. Statistical Analysis

Statistical analyses were performed in the XLSTAT program 2024.2.2 (1422). A paired *t*-test was used to determine significant differences between the observed results of the functional traits and antibacterial activity assays between the *L. plantarum* LB1_P46 and *L. casei* Shirota as probiotic control. An analysis of variance (ANOVA) with Dunnett’s post hoc test was performed to determine if the observed differences in the log CFU/mL in *L. plantarum* LB1_P46 and *L. casei* Shirota after 4 and 24 h of exposure to pH 3.5, 0.3% bile salts, and the combined condition pH 3.5 + 0.3% bile salt were significant by comparing the means of the assayed conditions for each strain with their respective controls. For the in vivo preventive anti-infective activity of *L. plantarum* LB1_P46 against *Salmonella* Typhimurium, to determine if the observed differences between the growth of *Salmonella* in the PBS condition and the administration of the LABs assayed were significant, an ANOVA with Tukey’s post hoc test was performed for the detected log CFU/g in the liver and spleen.

## 3. Results

### 3.1. Safety Traits of L. plantarum

The safety traits assayed for *L. plantarum* LB1_P46 were hemolytic activity and antibiotic resistance/susceptibility profiling. *L. plantarum* LB1_P46 resulted in negative hemolytic activity, whereas the profiling for antibiotic resistance/susceptibility is shown in Table 1. Results indicating sensitivity and intermediate sensitivity to PEP (<0.24 mg/L), PES, AMOP, ERY, QDA, CLI, LVX, CMP, and TSU antibiotics, but resistance to >0.125 mg/L of PEP, CTXP, TET, and VAN. The resulting MAR index for the strain LB1_P46 = 0.333 was calculated from the observed resistance to 4 of the 12 assayed antibiotics.

We searched the genome of strain LB1_P46 for possible corresponding genes encoding resistance to the assayed antibiotics, which resulted in the genes shown in Table 2. 

### 3.2. Functional Traits and Cell Surface Properties

The *L. plantarum* strain LB1_P46 was assayed for cholesterol reduction and β-galactosidase activity as functional traits, and for the cell surface properties of hydrophobicity, auto-aggregation, and aggregation. The results of these assays are shown in Table 3. The strain LB1_P46 showed a significant (*p* < 0.05) increment in the reduction of cholesterol of 56.03%, an increment of 21.17% compared to the control probiotic bacteria. Microbial cholesterol degradation was proposed mainly through the enzymatic deconjugation of bile acids by a bile-salt hydrolase or cholesterol conversion into coprostanol by a cholesterol reductase activity [34]. The search for annotated proteins encoded in the chromosome related to cholesterol degradation led to the identification in the genome annotation by Prokka of two enzymes possibly involved in bile acid degradation: the bile acid 7-alpha dehydratase (complement 2,710,282..2,710,726) forms secondary bile acids via the 7-alpha-dehydroxylation of primary bile acids, and the 3-alpha-hydroxycholanate dehydrogenase (NADP(+)) (complement 1,739,730..1,740,449) involved in the modification of secondary bile acids to form 3-beta-bile acids (also known as iso-bile acids) via a 3-oxo intermediate [35].

The strain LB1_P46 showed significantly (*p* < 0.05) lower values for all cell surface characteristics assayed compared to the probiotic control strain (Table 3). For hydrophobicity in chloroform and hexane, the strain LB1_P46 showed 58.38% and 83.75%, respectively, compared to the observed values in the control strain in both solvents. Regarding auto-aggregation, the strain LB1_P46 showed a significant (*p* < 0.05) reduction (~33%) of the observed for the control strain. Finally, the percentage of coaggregation reduced significantly (*p* < 0.05) with pathogenic bacteria at 24 h compared to the values observed for the control strain: ~28% for EPEC, ~22% for *L. monocitogenes*, and ~23% for *Salmonella* Typhi. 

Regarding the β-galactosidase activity, the strain LB1_P46 was positive for this trait (Table 1), resulting in blue colonies growing on lactose medium plates supplemented with X-gal, indicating the capability to transport and ferment lactose despite the non-dairy origin of this LAB. Analysis of the genome region encoding this hydrolytic enzyme revealed the presence of a β-galactosidase enzyme (locus_tag RUO99_14335 protein_id WNW15683.1) and a β-galactosidase small subunit (locus_tag RUO99_14405, protein_id WNW15697.1) located in a chromosomal region of 20,099 bp (3,021,825–3,041,923 bp), including several genes encoding transporters, carbohydrate metabolism enzymes, two LacI family DNA-binding transcriptional regulators (locus_tag RU099_14340 and RU099_14380), and three genes encoding the respective PTS lactose/cellobiose transporter subunits II (locus_tag RUO99_05865 protein_id WNW16884.1, locus_tag RUO99_11285 protein_i WNW15120.1, and locus_tag RUO99_03815l protein_id WNW16520.1, respectively). 

Analysis of the genome annotation by Prokka led to the identification of two encoded lactose permease (LacS). Nevertheless, GenBank annotation indicated these two proteins as glycoside-pentoside-hexuronide (GPH): cation symporters (locus_tag RUO99_14330, protein_id WNW15682.1 and locus_tag RUO99_14415, protein_id WNW15699.1, complement 3,044,336..3,046,294, respectively). Protein-BLAST analysis identified a lactose permease with an identity of 99.69% and 99.54% of *L. plantarum*, respectively. Although it is reported that *L. plantarum* transports lactose by LacS, further evidence is required to determine the main role of LacS or the PTS lactose/cellobiose transporter in transporting this sugar.

### 3.3. In Vitro Assays

#### Bile Salt and Acid Resistance

Results of the exposure at 4 and 24 h of strain LB1_P46 to acid pH (3.5), 0.3% bile salts, and a combined stress condition of acidified MRS (pH 3.5) supplemented with 0.3% of bile salts at 37 °C are shown in Figure 1. For the strain LB1_P46, all the assayed conditions significantly reduced the log CFU/mL of the respective control condition at 4 and 24 h of exposure. The combined pH 3.5 + 0.3% bile salts at 24 h proved to be the most stressful condition assayed, resulting in a final log CFU/mL = 4.46 compared to the log CFU/mL in the control at 24 h = 9.07. The probiotic control showed no significant differences in survival at individual pH 3.3 and 0.3% bile salt exposure. Nevertheless, at 24 h of exposure, the probiotic strain looks more resistant to the combined pH 3.5 + 0.3% bile salts at 4 and 24 h.

Analysis in the sequenced genome of the strain LB1_P46led for the *bsh* and *clpL* genes encoding a bile salt hydrolase and a Clp ATPase involved in acid and bile tolerance, respectively, reported previously in *L. plantarum* isolated from Indian fermented foods [11], led to the identification of these genes: *bsh* (locus_tag RUO99_14595, protein_id WNW15732.1) encoding a choloylglycine hydrolase and *clpL* (locus_tag RUO99_14745, protein_id WNW15759.1) encoding an ATP-dependent protease ClpL.

Regarding acid resistance of LAB, DltA (*L. rhamnosus*) and DltB (f *L. reuteri*), with the putative function of d-anylation of LTA, were proposed and detected in two *L. plantarum* from Indian fermented foods as involved proteins with acid and defensin resistance [11]. For its part, the bacterial community analysis of the Chinese baijiu fermentation showed that *Lactobacillus* was the most dominant bacterium during fermentation, and acid resistance genes identified were *argR*, *aspA*, *ilvE*, *gshA*, *DnaK*, and *cfa*, associated with the survival of *Lactobacillus* in the late stages of fermentation with high contents of acid and ethanol [36]. All these genes and an acid shock protein were detected in the chromosomal sequence of the strain LB1_046 by the analysis of the Prokka and GenBank annotations (Table 4).

### 3.4. Antibacterial Assays

#### 3.4.1. In Vitro Antibacterial Assays

The antimicrobial activity assays indicated that *L. plantarum* LB1_P46 can significantly inhibit in vitro (*p* < 0.05) the growth of *Salmonella* Typhimurium, *E. coli*, *P. aeruginosa*, *L. monocytogenes*, *S. pyogenes*, *S. aureus*, *E. faecalis*, and *B. cereus* (Table 5, Appendix A); the strain of *L. casei* Shirota was used as the control probiotic strain for this assay (Appendix A).

The results showed bacteriostatic activity in the assays associated with cell-to-cell contact against all pathogenic bacteria assayed. This suggests that the observed antimicrobial activity is likely related to a cell-associated diffusing enzyme or peptide with antimicrobial activity. Cell-to-cell contact assays showed a higher inhibition of *L. plantarum* LB1_P46 against *Salmonella* Typhimurium and *L. monocitogenes* (15.5 mm and 27.2 mm, respectively). In contrast, a lower inhibition was observed against *P. aeruginosa* (6.3 mm) (Table 5, Appendix A). *L. casei* Shirota used as control probiotic bacteria showed antimicrobial activity against all pathogenic bacteria assayed, still, to a lesser extent than the observed inhibition zones of the strain LB1_P46, ranging to 16.6 mm against *B. cereus* to 4.9 mm to *P. aeruginosa* (Table 5, Appendix A). 

To identify the possible enzyme or peptide responsible for this antimicrobial activity of the strain LB1_P46, we performed an in silico mining for biosynthetic clusters encoding any bacteriocin or a possible RiPP in the antiSMASH 7.0 bacterial version database [37] and the BAGEL4 server [38]. Results indicated the presence of two chromosomal regions located from 392,458 to 414,826 bp in the genome of *L. plantarum* LB1_P46 encoding for a predicted bacteriocin biosynthetic cluster (392,458–403,458), including an enterocin_X_chain_beta (RUO99_01860), a putative bacteriocin immunity protein (RUO99_01875), two plantaricin J genes (RUO99_01880 and RUO99_01885), a plantaricin A (RUO99_01895), and a bacteriocin-production-related histidine kinase (RUO99_01900). The second region comprises a RiPP-like encoding region (420,459–414,826 bp), including two response regulators, PlnC and PlnD activators (RUO99_01905, and RUO99_01910, respectively), an immunity protein PlnI membrane-bound protease CAAX (RUO99_01920), the subunits E and F of the plantaricin EF (RUO99_01925 and RUO99_01930, respectively), a bacteriocin ABC-transporter, ATP binding and permease protein PlnG (RUO99_01935), and the accessory factor for ABC-transporter PlnH (RUO99_01940) (Figure 2). The mined genomic structure corresponding to the RiPP-like region showed the typical structure of an operon encoding class II bacteriocins [39,40].

Other enzymes with possible antimicrobial activity encoded in the chromosome of *L. plantarum* LB1_P46 are five enzymes involved in the peptidoglycan catabolic process, four with lysozyme activity (GH25 family lysozyme, locus_tag, protein ID): RUO99_04720 WNW16683.1, RUO99_05000 WNW16738.1, RUO99_07360 WNW17169.1, RUO99_04720 WNW16683.1, RUO99_11435 WNW17283.1; and one enzyme with N-acetylmuramoyl-L-alanine amidase (RUO99_08270 WNW14576.1).

#### 3.4.2. In Vivo Anti-Infective Activity of *L. plantarum* LB1_P46 Isolate against *Salmonella* Typhimurium

The anti-infective effect of the oral administration of *L. plantarum* LB1_P46 in BALB/c male mice of age 8–9 weeks against *Salmonella* Typhimurium str^r^ was determined by quantification of the log CFU/mL of *Salmonella* in dissected liver and spleen of infected mice compared. All the experimental groups were monitored for weight to discard a possible gain associated with the LAB administration, resulting in no weight gain. The mice of the LB1_P46 group were bright-eyed and alert, had a smooth coat with a sheen, responded to stimuli, and showed interest in their environment. In contrast, the PBS mice group showed visible signs of infection after administering *Salmonella*, being hunched over and exhibiting lethargy, little interest in the environment, and fur clumping.

The administration of the strain LB1_P46 to the BALB/c mice before the infection with the strain *Salmonella* Typhimurium L1334 resulted in a significant reduction in the log CFU/g tissue (liver and spleen) of *Salmonella* than in the non-LAB supplemented group (Figure 3). The observed log CFU in the PBS group was 6.08 for the liver. The observed values for the LB1_P46 group were 3.92 and 5.02 for the *L casei* Shirota group. The PBS group showed a log CFU of 4.86 for the spleen, reducing to 3.02 and 3.81 for the LB1_46 and the *L. casei* Shirota groups, respectively. These results showed that administering *L. plantarum* LB1_P46 to BALB/c male mice confers an effective preventive anti-infective effect against the strain of *Salmonella* assayed compared to that observed with the commercial probiotic strain.

Several genes have been associated with the in vivo antipathogenic effect of *L. plantarum*, including *luxS* associated with the putative production of AI-2 and AI-3 conferring an autoinduction ability, *dltB* and *dltD* with the putative function of d-anylation of LTA, conferring an anti-inflammatory potential in a murine model of colitis (*dltB*) and resistance to human β defensin-2 [11], and seven unspecified genes related to infection for *Salmonella*, tuberculosis, and legionellosis [30]. The analysis of the genome sequence of the strain LB1_P46 led to identifying *luxS* (annotated as *S*-ribosylhomocysteine lyase in the GenBank annotation but LuxS by Prokka) locus_tag RUO99_03160, protein_id WNW16399.1, *dltD* (D-alanyl-lipoteichoic acid biosynthesis protein DltD), locus_tag RUO99_08340, protein_id WNW14589.1, and *dltB* (product D-alanyl-lipoteichoic acid biosynthesis protein DltB), locus_tag RUO99_08350, protein_id WNW14591.1.

## 4. Discussion

*L. plantarum* LB1_P46 showed a MAR index = 0.33, resulting from resistance to the PEP, CTXP, and VAN antibiotics. Several strains of *L. plantarum* isolated from traditional fermented sources have been demonstrated as resistant LAB for several antibiotics, mainly synthetic quinolones such as nalidixic acid and ciprofloxacin [22,29], targeted mainly against pathogenic Gram-negative bacteria [41], and resistance to the glycopeptide vancomycin [42]. Resistance to vancomycin was proposed to be caused by the presence of peptidoglycan precursors terminating in the depsipeptide D-Ala-D-Lac observed in *vanA* and *vanB* resistance classes to vancomycin observed in *Enterococci*, resulting in the weak affinity of the antibiotic to the LAB’s cell walls [43]. Other *L. plantarum* isolated from fermented sources exhibited MAR indexes higher than 0.2, indicating multiple antibiotic-resistant bacteria [29]. Nevertheless, four *L. plantarum* isolated from pulque beverages from the States of Puebla and Oaxaca in Mexico showed MAR < 0.2 indexes (calculated from published values in Ruiz-Ramírez et al.) [22].

The analysis of the sequence of the complete chromosome of the strain LB1_P46 led to identifying the encoded resistance proteins for PEP and possibly for CTXP (a serine hydrolase member of the GO:0030655—beta-lactam antibiotic catabolic process) [44,45]; TET (TetM/TetW/TetO/TetS family tetracycline resistance ribosomal protection protein, tetracycline resistance MFS efflux pump, and ABC multidrug transporters) [46,47,48]; and VAN (VanZ family protein), a member of the VanA glycopeptide resistance gene cluster, reported to confer resistance to lipoglycopeptide antibiotics independent of activity of the encoded proteins activity of the *vanHAX* genes [49]. The ABC transporter permease/daunorubicin/doxorubicin resistance ABC transporter permease proteins DrrA and DrrB are reported as involved in the resistance to these antibiotics.

Other potential MFS, MDR, and ABC multidrug efflux transporters were also detected, and, remarkably, 21 genes encoded the TetR/AcrR family transcriptional regulator and two genes encoded the TetR-like C-terminal domain-containing protein (Table 2). TetR was reported as a sensor of the cellular environment and its dynamics. Genes under the control of TetR are involved in functions like antibiotic production, osmotic stress, and multidrug resistance, including multidrug efflux pumps, pathogenesis, and metabolic modulation processes. Remarkably, the first TetR sensor was reported to regulate the tetracycline efflux pump, resulting in resistance to this antibiotic [50]. Our antibiotic resistance assay did not include the quinolone antibiotics ciprofloxacin and ofloxacin; nevertheless, *Lactobacillus* resistance to these antibiotics is a common trait in this LAB. Antibiotic profiling in other *L. plantarum* isolated from Mexican pulque [22] and boza from Turkey [29] report resistance to quinolone antibiotics. Proposed resistance mechanisms to quinolones include chromosomal mutations in the target enzymes, DNA gyrase (*gyrAB*) and DNA topoisomerase IV subunit V (*parCE*), down-regulation of outer membrane porin coupled with increases in active drug efflux, and the possible acquisition of transmissible quinolone-resistance genes [22,51].

Potential probiotic strains isolated from traditional fermented sources resistant to several antibiotics help control intestinal infections due to probiotics’ intrinsic preventive and therapeutic properties. Still, their associated antibiotic-resistant traits could be considered helpful in promoting faster recovery of patients by promoting the rapid establishment of desirable microbiota if resistant probiotic LAB are administered simultaneously with antibiotics [29]. Nevertheless, as *Lactobacillus* (and *Lactiplantibacillus*) species are generally considered nonpathogenic bacteria, many strains of these LABs are used to produce diverse foods and products for humans and animals. Still, multidrug resistance observed for several LABs exceeds the resistance levels recommended by the EFSA [29,42].

A relevant characteristic of the strain LB1_P46 is its capability to grow at 37 °C under stress-challenged conditions, compared to the optimal growth environmental temperature of this bacteria in its natural environment in the sap in maguey species and during the pulque fermentation process. Acid tolerance (pH 3.5), bile tolerance, and combined acid and bile tolerance were assayed for strain LB1_P46 at 37 °C. Acid resistance in lactobacilli involved in traditional Chinese fermented foods was associated with the activity of proteins ArgR, Cfa, AspA, IlvE, DNaK, and GshAB [36]. All these genes were encoded in the genome of the strain LB1_P46 (Table 4). The proposed role of acid-resistance genes involves the transcriptional regulator ArgR of the arginine deaminase pathway (ADI), which was proposed to be under antirepression control by the small RNA S042 of *argR* in *Lactococcus lactis*. The expression of the ADI pathway results in the intracellular production of ammonia, which neutralizes intracellular protons and increases intracellular pH [36,52]. Cyclopropane fatty acyl phospholipid synthase (CFA), encoded by *cfa*, is an enzyme that catalyzes a modification of the acyl chains of phospholipid bilayers through methylenation of unsaturated fatty acyl chains to their cyclopropane derivatives. Mutants of *E. coli* in *cfa* and *cfa-rpoS* showed a reduced resistance mutant, showing enhanced acid sensitivity. In *E. coli*, the cytoplasmic membrane CFA protects the cells against several environmental stressors, including acid pH [36,53,54]. The amino acid-dependent system involving the deamination of aspartate by aspartate ammonia-lyase (AspA) results in the reversible formation of fumaric acid and ammonium ion, which was proposed to increase intracellular pH. No pH changes in a Δ*aspA* mutant was observed [36,55,56].

The branched-chain amino acid aminotransferase (IlvE) is involved in the biosynthesis and degradation of branched-chain amino acids, such as in the production of branched-chain fatty acids in *Streptococcus mutants*. Δ*ilvE* mutants resulted in a decreased acid tolerance, and the transcriptional analysis of the *ilvE* regulator was upregulated during growth under acid pH [36,57]. The molecular chaperone DnaK (*dnaK*) was reported as involved in the extended response to acid pH in *Streptococcus infantarius* isolated from traditional Mexican fermented pozol. Upregulation of *dnaK* was observed during exposure to pH 3.6 and 4.0 [58]. Its upregulation was demonstrated to be associated with the thermo-acidophilic stress of *Alicyclobacillus acidoterrestris*, a Gram-positive bacteria isolated from orchard soil, spoiled fruit juices, and soils closer to volcanoes and hot springs, growing in acid pHs from 2.0 to 6.0 [59].

Regarding GshAB, LAB developed a glutathione (GSH)-mediated acid-tolerance response. However, several reports indicate the absence of GshA or GshAB in the synthesis of GSH in several LAB like *L. salivarius* or *L. plantarum*, resolving availability by transport of GSH from the extracellular medium [60]. The GSH-mediated acid response involves using glutathione to avoid rapid acidification of the intracellular pH [61]. The presence of two GshAB in the strain LB1_P46 requires deep analysis to demonstrate its possible role in the GSH biosynthesis and the possible participation of GSH in a mechanism of acid resistance as proposed for other LAB.

Regarding bile tolerance, the product of the *bsh* gene (locus_tag RUO99_14595) is a choloylglycine hydrolase (a bile salt hydrolase) according to GenBank annotation of the genome of the strain LB1_P46. This enzyme cleaves the peptide bond of the bile acids, removing the amino acid residing in the steroid core, and the unconjugated resulting bile acids are then precipitated at acid pH [62]. The ClpL protein was reported as involved in acid and bile tolerance in *L. plantarum* isolates from Indian fermented foods [11] and was detected in the genome of the LB1_P46 strain. Nevertheless, the ClpL protein was reported in several LAB, including the probiotic strain of *L. plantarum* IS-10506 as expressed in response to heat shock, but is also considered a desirable trait for surviving probiotic bacteria during the manufacturing and administration process [30]. Another enzyme possibly involved in bile tolerance in the strain LB1_P46 is the bile acid 7-alpha dehydratase (locus_tag RUO99_12795), an enzyme reported to catalyze the fourth step in the bile acid 7-alpha-dehydroxylation pathway, resulting in secondary bile acids via the 7-alpha-dehydroxylation of primary bile acids, a pathway carried out by intestinal anaerobic bacteria [35].

Cell surface characteristics such as hydrophobicity, auto-aggregation, and coaggregation are crucial for probiotic bacteria. Obtained values for cell surface characteristics assayed for the strain LB1_P46 are in the range of previous LAB isolated from pulque beverage analyzed for similar traits [22]. Hydrophobicity correlates with the adhesion ability of probiotic bacteria to the host’s epithelial cells and is considered an essential characteristic of probiotics’ cell surface properties. Coaggregation is considered an effective host defensive mechanism against pathogenic microorganisms in the host’s gastrointestinal system, whereas auto-aggregation is a desirable precondition for biofilm development [63,64]. The operon *dltABCD* was detected in the chromosome of the strain LB1_P46 in the region of 1,764,114 to 1,762,640. It was reported in several LAB encoding the pathway of esterification of lipoteichoic acid (TLA) by D-alanine; resulting esters are reported to play an essential role in controlling the net anionic charge for the poly (GroP) moiety of LTA [65]. This operon has been proposed to play a pleiotropic role in the properties of the cell surface. Inactivation of the genes in the operon has resulted in the loss of integrative coaggregation (*dltA* mutant), loss of acid tolerance (*dltC* mutant), and increased cellular length and increased cellular susceptibility of CTAB and chlorhexidine (*dltD* mutant) associated with a decreased in D-alanylation of LTA [65,66].

A remarkably functional trait of the strain LB1_P46 is its capability to transport lactose, probably by the identified lactose permease (LacS), and to hydrolyze the disaccharide as a carbon source by the action of the β-galactosidase enzyme. We previously observed this bacteria’s ability to grow in skim milk to a final 9.14 log CFU/mL at 37 °C for 24 h [67], suggesting the potential use of this strain in the elaboration of dairy probiotic preparations.

The capability to produce bacteriocins confers a competitive advantage to probiotic bacteria while colonizing the host’s intestinal tract. *L. plantarum* produces diverse bacteriocins (plantaricins) that have acquired great importance as bio preservatives in the preparation of diverse dairy, meat, and fish products, and for their potential use in treating infective diseases, including a protective role in urinary tract infections [28]. We identified three predicted bacteriocins in the chromosomal sequence of the strain LB1_P46: two plantaricin J, one plantaricin A, and the two subunits of the plantaricin EF, including immunity proteins and transporters. Four lysozyme enzymes were also identified, including one member of the GH25 lysozyme family and one enzyme with N-acetylmuramoyl-L-alanine amidase. Interestingly, the lysozyme family GH25 exhibits both β-1,4-*N*-acetyl- and β-1,4-*N*,*6-O*-diacetylmuramidase activities and is structurally unrelated to the GH22-24, GH73, and GH108 lysozyme families [38]. All identified plantaricins and cell wall hydrolytic enzymes could confer a competitive advantage during the development of this bacteria in the fermentation process for pulque production.

*Lactobacillus* isolates have antagonistic activity against pathogenic bacteria, including microorganisms like *Staphylococcus aureus*, *Enterococcus faecalis*, *Klebsiella pneumonia*, *Pseudomonas aeruginosa*, *E. coli*, *Salmonella* Typhi, and *Shigella* sp. [68], whereas the in vivo preventive and anti-infective of probiotic *Lactobacillus* have been assayed against *Salmonella*. Typhimurium [69]. Our results showed the in vitro antagonistic effect of *L. plantarum* LB1_P46 against *Salmonella* Typhimurium, *E. coli*, *P. aeruginosa*, *L. monocytogenes*, *S. pyogenes*, *S. aureus*, *E. faecalis*, and *B. cereus*; and the anti-infective treatment of the oral delivery of the strain of *L. plantarum* LB1_P46 against the infection of *Salmonella* Typhimurium in a BALB/c male mouse model, resulting in a relevant probiotic trait of this LAB isolated from pulque beverage. The oral administration of probiotic bacteria has been used to reduce the infection of pathogenic bacteria involving immune-enhancing properties by releasing anti-inflammatory cytokines within the gut by impacting various types of immune cells, including dendritic cells, monocytes, natural killer cells, macrophages, lymphocytes, and epithelial cells simultaneously (reviewed in [70]). Several genes have been identified and proposed as activating the immune response during the in vivo anti-infective treatment of probiotic *Lactobacilllus* against pathogenic bacteria, including *luxS* and *dltBD* [11,65,66]. DltB and DltD proteins involved in the D-anylation of LTA and encoded in the *dltABCD* operon (encoded in the chromosome of the strain LB1_P46) have been associated with anti-inflammatory activity. Mutants in *dltD* exhibited reduced anti-inflammatory potential. Mutants of *L. plantarum* in *dltB* showed significantly reduced immunomodulatory activity compared to its parental strain both in in vitro and in vivo models. In contrast, mutants in *dltD* in *L. rhamnosus* GG, cytokine stimulation of human intestinal epithelial cells, and peripheral blood mononuclear cells were not significantly altered [66]. Additionally, the LTA glycolipid anchor and the length of the Gro-P backbone have been reported to have immunostimulatory potential in other Gram-positive bacteria [66], resulting in interest in the role of the operon *dltABCD* and the structure of LTA in the *L. plantarum* strain LB1_P46.

## 5. Conclusions

The present study investigates the probiotic characteristics of *L. plantarum* LB1_P46 isolated from traditional Mexican fermented pulque beverage, including several functional traits, such as the determination of antagonistic in vitro activity against several pathogenic bacteria, and in vivo anti-infective effect against the infection of *Salmonella* Typhimurium in a murine model. Along with the functional and antimicrobial properties assayed, we identified several probiotic genes, operons, and biosynthetic clusters in the genome of this bacteria that are probably responsible for the observed probiotic characteristics, indicating that *L. plantarum* LB1_P46 is a potential probiotic candidate that can be used in non-dairy and dairy functional products. Additionally, our results provide scientific evidence supporting several beneficial effects of consuming traditional pulque beverage.

## Figures and Tables

**Figure 1 microorganisms-12-01652-f001:**
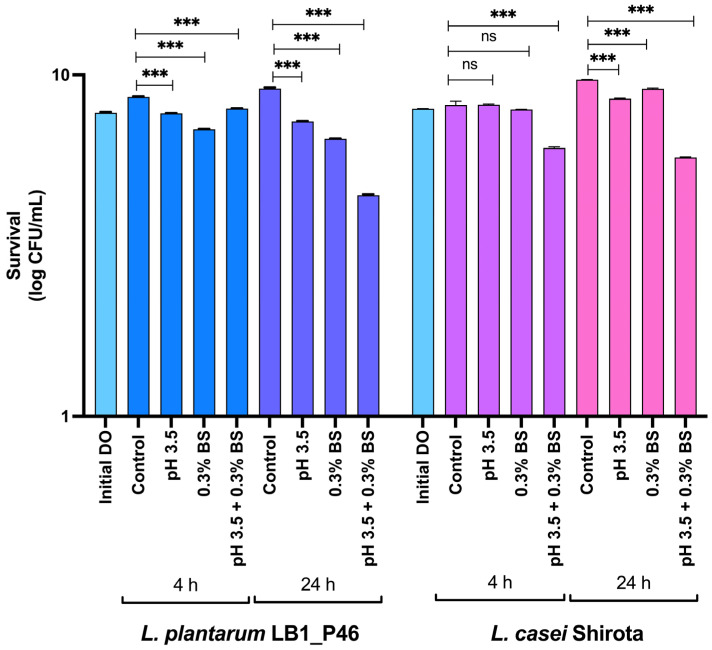
The survival of *L. plantarum* LB1_P46 and *L. casei* Shirota to pH 3.5 and 0.3% bile salt exposition at 4 and 24 h. The initial DO_600 nm_ = 0.2 in the inoculum. All data are shown as the average and SD values for each determination. BS, bile salts. Significant differences in one-way ANOVA with Dunnett’s post hoc test between controls and assayed conditions are indicated: *** *p* < 0.0001; ns, non-significant differences.

**Figure 2 microorganisms-12-01652-f002:**
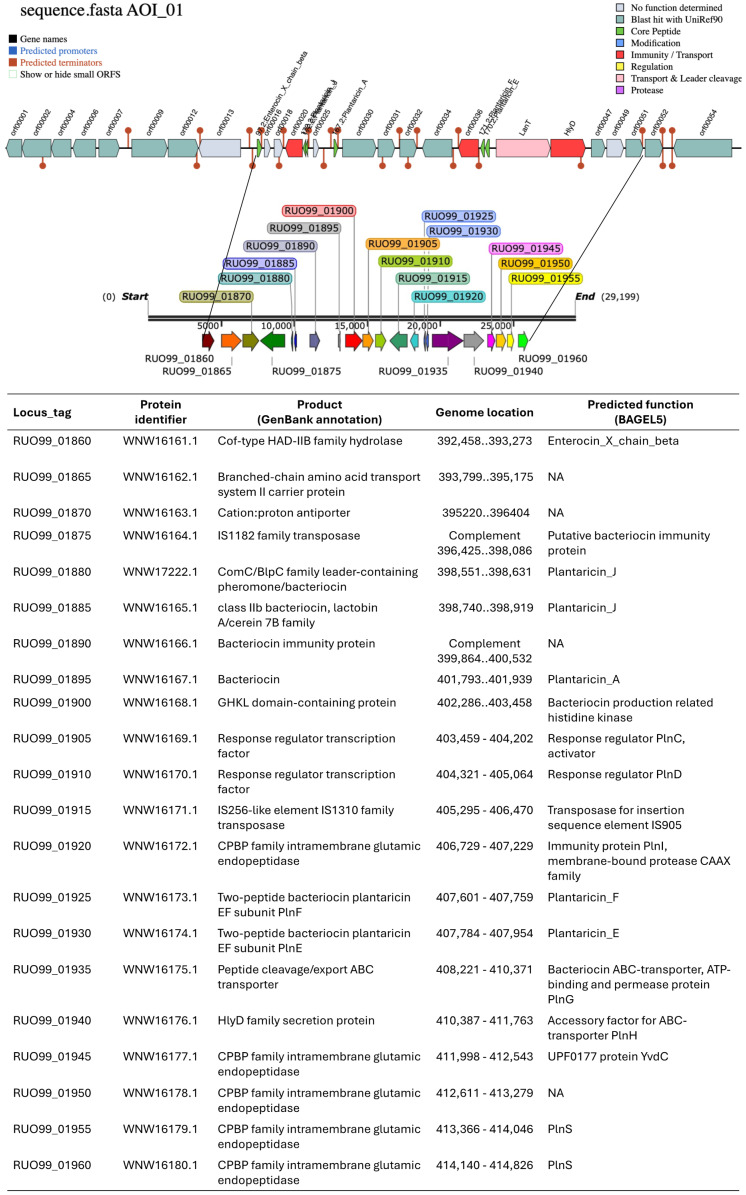
The chromosomal region of *L. plantarum* LB1_P46 encodes a bacteriocin-encoding gene cluster and a RiPP-like region as predicted by the antiSMASH 7.0 bacterial version database [37] and the BAGEL4 server [38]. The upper section of the figure showing the genomic location of the mined region bacteriocin-RiPP-like regions (29,199 bp) corresponds to the output analysis in the BAGEL4 server of the entire chromosome of *L. plantarum* LB1_P46. Detailed locus_tag assignment of the genes in the bacteriocin—RiPP-like encoding region (392,458–414,826 bp) was generated in the SnapGene 7.2.1 program (San Diego, USA, GSL Biotech LLC). Detailed information on the corresponding locus_tag, protein ID, encoded product, genomic location, and predicted function was retrieved from the antiSMASH and BAGEL4 analysis output. NA, not assigned function in the BAGEL4 output results.

**Figure 3 microorganisms-12-01652-f003:**
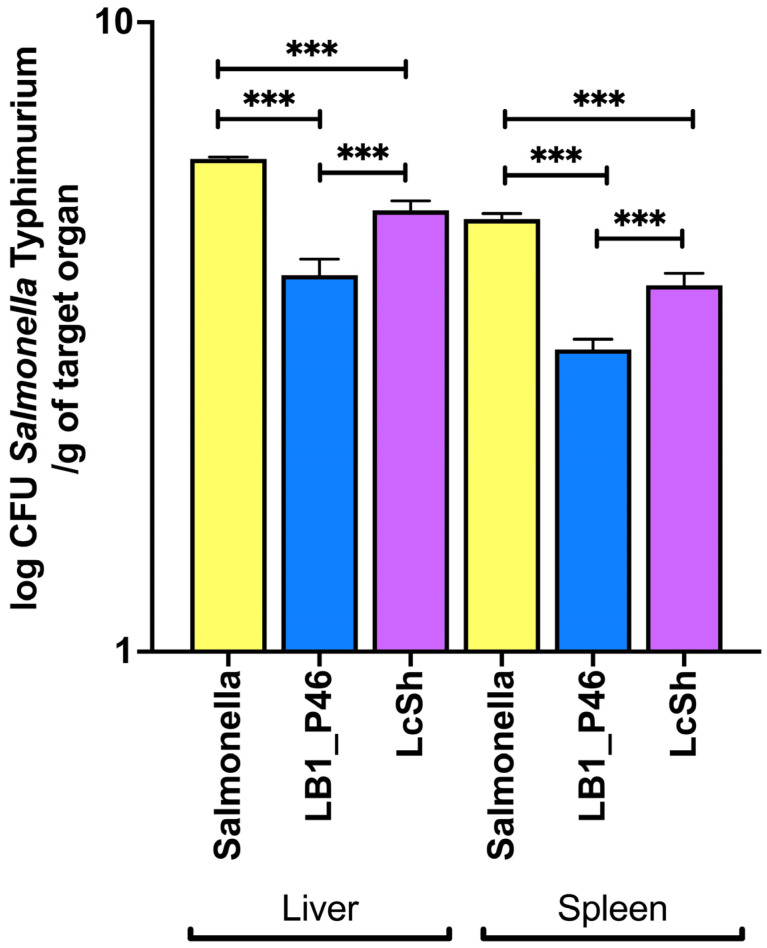
Anti-infective effect of *L. plantarum* LB1_P46 on *Salmonella* Typhimurium str^r^ in mouse liver and spleen (blue bars). The yellow bars correspond to the infection (PBS) group without LAB administration, and the lavender bars correspond to the probiotic group with *L. casei* Shirota. Values are the average and SD of log CFU/g liver and spleen samples obtained from ten mice per group. Significant differences in one-way ANOVA with Tukey’s post hoc test between assayed groups are indicated: *** *p* < 0.0001.

**Table 1 microorganisms-12-01652-t001:** Antibiotic susceptibility profile of *L. plantarum* LB1_P46.

Antibiotic	Concentration (mg/L)	Antibiotic Susceptibility
Penicillin (PEP)	0.031–0.5	R
	0.063–1.0	R
	0.125–2.0	R
	0.25–4.0	I
Penicillin–Streptomycin (PES)	0.12–2.0	I
Amoxicillin (AMOP)	2.0–4.0	S
Cefotaxime (CTXP)	0.5–2.0	R
	0.5–1.0	R
Erythromycin (ERY)	0.25	S
Quinupristin/dalfopristin (QDA)	1.0	S
Clindamycin (CLI)	0.25	S
Tetracycline (TET)	2.0	R
Levofloxacin (LVX)	2.0–4.0	I
Chloramphenicol (CMP)	4.0	S
Vancomycin (VAN)	1.0	R
Cotrimoxazole (TSU) (trimetoprima y sulfametoxazol)	0.5/9.5–2/38	S

R, resistant; S, sensitive; I, intermediate sensitivity.

**Table 2 microorganisms-12-01652-t002:** Antibiotic resistance encoding genes and products in the complete genome of *L. plantarum* LB1_P46.

Gene/locus_tag ^1^	Encoded Protein	Protein_id ^1^
RUO99_00065	MFS transporter ^1^, Multidrug resistance protein MdtG ^2^	WNW15827.1
RUO99_00280	TetR/AcrR family transcriptional regulator ^1^	WNW15867.1
RUO99_00405	TetM/TetW/TetO/TetS family tetracycline resistance ribosomal protection protein ^1^, Tetracycline resistance protein TetO ^2^	WNW15891.1
RUO99_00755	TetR/AcrR family transcriptional regulator ^1^	WNW15960.1
RUO99_01090	Small multidrug resistance protein ^1^	WNW16015.1
RUO99_01335	TetR/AcrR family transcriptional regulator ^1^	WNW16064.1
RUO99_01430	TetR-like C-terminal domain-containing protein ^1^	WNW16081.1
RUO99_02075	Multidrug efflux MFS transporter ^1^	WNW16202.1
RUO99_03140	Tetracycline resistance M.F.S. efflux pump ^1^	WNW16395.1
RUO99_03385	TetR/AcrR family transcriptional regulator ^1^	WNW16441.1
RUO99_03440	VanZ family protein ^1^	WNW16450.1
RUO99_03670	TetR/AcrR family transcriptional regulator ^1^	WNW16492.1
RUO99_03685 (pseudo)	Multidrug efflux SMR transporter ^1^	
RUO99_04150	TetR/AcrR family transcriptional regulator ^1^	WNW16574.1
RUO99_04695	TetR/AcrR family transcriptional regulator ^1^	WNW16678.1
RUO99_05535	MDR family MFS transporter ^1^, Multidrug resistance protein 3 ^2^	WNW16825.1
RUO99_06080	TetR family transcriptional regulator ^1^	WNW16925.1
RUO99_06590	TetR/AcrR family transcriptional regulator ^1^	WNW17024.1
RUO99_06840	TetR/AcrR family transcriptional regulator ^1^	WNW17073.1
RUO99_07040 (pseudo)	TetR/AcrR family transcriptional regulator ^1^	
RUO99_07045	MFS transporter/Putative multidrug resistance protein MdtD ^1^	WNW17111.1
RUO99_07060	TetR family transcriptional regulator ^1^	WNW17114.1
RUO99_07745	MFS transporter ^1^, Multidrug resistance protein 3 ^2^	WNW14477.1
RUO99_07940	ABC transporter permease ^1^, Daunorubicin/doxorubicin resistance ABC transporter permease protein DrrB ^2^	WNW14514.1
RUO99_07945	Daunorubicin resistance protein DrrA family ABC transporter ATP-binding protein ^1^	WNW14515.1
RUO99_09860	Serine hydrolase (GO:0030655—beta-lactam antibiotic catabolic process) ^1^	WNW14850.1
RUO99_10165	ABC transporter ATP-binding protein ^1^, Multidrug resistance ABC transporter ATP-binding/permease protein BmrA ^2^	WNW14908.1
RUO99_10430	VanZ family protein ^1^	WNW14958.1
RUO99_11265	ABC transporter ATP-binding protein ^1^, Linearmycin resistance ATP-binding protein LnrL ^2^	WNW15116.1
RUO99_11750	A.B.C. transporter ATP-binding protein ^1^, Putative multidrug resistance A.B.C. transporter ATP-binding/permease protein YheH ^2^	WNW15206.1
RUO99_11750	A.B.C. transporter ATP-binding protein ^1^, Putative multidrug resistance A.B.C. transporter ATP-binding/permease protein YheI ^2^	WNW15206.1
RUO99_11780	TetR/AcrR family transcriptional regulator ^1^	WNW15212.1
RUO99_12010	TetR-like C-terminal domain-containing protein ^1^	WNW15254.1
RUO99_12025	ABC transporter ATP-binding protein ^1^, Multidrug resistance A.B.C. transporter ATP-binding and permease protein ^2^	WNW15256.1
RUO99_12525	TetR/AcrR family transcriptional regulator ^1^	WNW15353.1
RUO99_12560	TetR/AcrR family transcriptional regulator ^1^	WNW15360.1
RUO99_12610	TetR/AcrR family transcriptional regulator ^1^	WNW17292.1
RUO99_12660	TetR/AcrR family transcriptional regulator ^1^	WNW15377.1
RUO99_12810	TetR/AcrR family transcriptional regulator ^1^	WNW15405.1
RUO99_13645	Multidrug efflux SMR transporter ^1^	WNW17299.1
RUO99_13760	VanZ family protein ^1^	WNW15579.1
RUO99_14075	TetR family transcriptional regulator ^1^	WNW15636.1
RUO99_14165	TetR/AcrR family transcriptional regulator ^1^	WNW15651.1
RUO99_14295	TetR/AcrR family transcriptional regulator ^1^	WNW15675.1

^1^ The NCBI Prokaryotic Genome Annotation Pipeline (P.G.A.P.), ^2^ Prokka annotation pipeline [33].

**Table 3 microorganisms-12-01652-t003:** Functional traits and cell properties of *L. plantarum* LB1_P46.

Assay	*L. plantarum* LB1_P46	*L. casei* Shirota
β-galactosidase activity	+	+
Assayed trait ^1^		
Cholesterol assimilation (%)	56.03 ± 0.96 b	46.21 ± 0.27
Hydrophobicity (%)		
Chloroform	37.65 ± 1.08 a	64.49 ± 0.43
Hexane	29.13 ± 3.14 NS	34.78 ± 0.27
Auto-aggregation (%)	20.67 ± 2.21 b	30.81 ± 1.56
Coaggregation (%)		
EPEC	39.12 ± 0.98 b	54.34 ± 1.04
*L. monocitogenes*	30.30 ± 1.24 c	38.86 ± 1.78
*Salmonella* Typhi	56.13 ± 1.71 b	73.17 ± 1.96

^1^ All data for functional assays and cell properties are shown in the average obtained from three independent experiments and SD values. For the β-galactosidase assay, each LAB was streaked in quadruplicate in lactose medium plates, and the resultant blue-colony phenotype was registered as positive. The letters correspond to the paired *t*-test results, indicating significant differences at *p* < 0.05 between the strain LB1_P46 and *L. casei* Shirota as probiotic control. *** *p* < 0.0001, ** *p* < 0.01, * *p* < 0.05. a = ***, b = **, c = *, NS, non-significant differences.

**Table 4 microorganisms-12-01652-t004:** Genes proposed as being involved in pH and bile salt resistance identified in the genome of *L. plantarum* LB1_P46.

Gene/locus_tag ^1^	Gene ^1,2^	Encoded Protein	Protein_id ^1^
RUO99_00625	NA ^1,2^	Hsp20/alpha-crystallin family protein ^1^, acid shock protein ^2^	WNW15934.1
RUO99_05920	*argR*^1^, argR_1 ^2^	Arginine repressor ^1,2^	WNW16894.1
RUO99_07080	NA ^1^, cfa_2 ^2^	Cyclopropane-fatty-acyl-phospholipid synthase family protein ^1,2^	WNW17118.1
RUO99_11515	*aspA* ^2^	Aspartate ammonia-lyase ^1,2^	WNW15160.1
RUO99_10045	NA ^1^, *ilvE* ^2^	Branched-chain amino acid aminotransferase ^1,2^	WNW14887.1
RUO99_12925	NA ^1^, cfa_2 ^2^	Cyclopropane-fatty-acyl-phospholipid synthase family protein ^1,2^	WNW15428.1
RUO99_08375	*dnaK* ^1,2^	Molecular chaperone DnaK ^1,2^	WNW14596.1
RUO99_09755	gshAB_2 ^2^	Glutamate-cysteine ligase ^1^, glutathione biosynthesis bifunctional protein GshAB ^2^	WNW14833.1
RUO99_09835	NA ^1^, gshAB_1 ^2^	Bifunctional glutamate--cysteine ligase GshA/glutathione synthetase GshB ^1^, glutathione biosynthesis bifunctional protein GshAB ^2^	WNW14847.1

^1^ The NCBI Prokaryotic Genome Annotation Pipeline (PGAP), ^2^ Prokka annotation pipeline; NA, Not assigned.

**Table 5 microorganisms-12-01652-t005:** Antimicrobial activity of *L. plantarum* LB1_P46 against several Gram-positive and Gram-negative pathogens.

Pathogen	*L. plantarum* LB1_p46	*L. casei* Shirota
	Growth Inhibition Zone (mm) ^1^	
*Salmonella* Typhimurium ATCC14028	17.5 ± 1.8 a	7.4 ± 0.4
*E. coli* 1129	11.2 ± 0 a	7.1 ± 0.0
*P. aeruginosa* ATCC27853	6.3 ± 0.6 a	4.9 ± 0.4
*L monocytogenes* CFQ-B-103	27.2 ± 0.6 a	9.6 ± 1.2
*S. pyogenes* CFQ-B-218	13.8 ± 0.9 a	9.9 ± 0.4
*S. aureus* ATCC6538	13.7 ± 0.5 a	9.9 ± 0.4
*E. faecalis* CFQ-B-	14.0 ± 0.5 b	13.3 ± 0.5
*B. cereus* CFQ-B-230	13.5 ± 1.4 a	16.6 ± 0.4

^1^ The value corresponds to the average and SD values of eight inhibition zones per assayed pathogen. Appendix A show the plate assays for *L. plantarum* LB1_P46 and *L. casei* Shirota against pathogenic bacteria. The letters correspond to the paired t-test results indicating significant differences at *p* < 0.05 between the strain LB1_P46 and *L. casei* Shirota as probiotic control. a = ***, b = *. *** *p* < 0.0001, * *p* < 0.05

## Data Availability

The raw data supporting the conclusions of this article will be made available by the authors upon request.

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
