# Peer review of "Analysis of the Probiotic Potential of Lactiplantibacillus plantarum LB1_P46 Isolated from the Mexican Fermented Pulque Beverage: A Functional and Genomic Analysis"

_microorganisms, 2024, doi:10.3390/microorganisms12081652_

Round 1

Reviewer 1 Report

Comments and Suggestions for Authors

This research evaluated the probiotic attributes of the L. plantarum LB1_P46 strain isolated from a pre-Hispanic traditional Mexican fermented pulque beverage. The study included an analysis of the antagonistic activity against various pathogenic bacteria in vitro,  the anti-infective effects against the infection of Salmonella in a BALB/c mouse model, identification of some probiotic genes, operons, and biosynthetic clusters in the genome of this bacterium.

While the study touches on an exciting and relevant topic, significant methodological and conceptual issues must be addressed to establish credibility and reliability. For example,  the number of samples, how the sample was treated before the beginning of the experiment, and the number of replications were not mentioned.

Questionable statistical rigor: there is no mention of the statistical analysis used to analyze the data and support the conclusions drawn from both in vitro and in vivo experiments. Proper statistical treatment is crucial to ensure that the observed effects are not due to chance and to lend credibility to the results.

Addressing these critical issues is essential for research to be considered a credible contribution to its field.

Author Response

This research evaluated the probiotic attributes of the L. plantarum LB1_P46 strain isolated from a pre-Hispanic traditional Mexican fermented pulque beverage. The study included an analysis of the antagonistic activity against various pathogenic bacteria in vitro,  the anti-infective effects against the infection of Salmonella in a BALB/c mouse model, identification of some probiotic genes, operons, and biosynthetic clusters in the genome of this bacterium.

Comments 1: While the study touches on an exciting and relevant topic, significant methodological and conceptual issues must be addressed to establish credibility and reliability. For example, the number of samples, how the sample was treated before the beginning of the experiment, and the number of replications were not mentioned.

Response 1: Dear reviewer, thank you for your valuable concerns. In the new version of the manuscript, we included a description of the number of replicates for each run experiment in the corresponding subsections in the Material and Methos and Results sections. Additionally, we included a brief description in the corresponding figure legend or foot table describing the statistics for the data shown. Find these modifications in the content of the new version of this manuscript typed in red text:

  1. Subsection 2.2.1 Hemolytic activity, line 144.
  2. Subsection 2.3.2 β-galactosidase activity, line 177.
  3. Subsection 2.4.3 Coagregattion, lines 215-216.
  4. Subsection 2.5.1 Bile and acid resistance, line 230..
  5. Subsection 2.5.2 Antibacterial assays against pathogenic bacteria, lines 237 – 238.
  6. Subsection 2.5.3 In vivo preventive anti-infective activity of…, lines 251-253. This subsection includes a new paragraph describing the statistical analysis performed to determine whether the observed differences from this experiment were significant, lines 270-274.
  7. Table 3, foot table, lines 320-323.
  8. Figure 1 legend, lines 361-362.
  9. Table 5, foot table, lines 395.
  10. Subsection 3.4.1, lines 457 and 459.
  11. Figure 3, figure legend, lines 480-481.

Finally, we checked all the methods describing the performed experiments. We considered that we described the proper sample preparation and, with the newest modifications, the number of replicates assayed.

Comments 2: Questionable statistical rigor: there is no mention of the statistical analysis used to analyze the data and support the conclusions drawn from both in vitro and in vivo experiments. Proper statistical treatment is crucial to ensure that the observed effects are not due to chance and to lend credibility to the results.

Response 2: Dear reviewer, thank you for your valuable concerns. As explained in the previous reply, we described the number of samples and replicates and the corresponding statistical analysis and data treatment for all experiments included in this contribution. For details for each section, please review the above-numbered reply.

Comments 3: Addressing these critical issues is essential for research to be considered a credible contribution to its field.

Response 3. Dear reviewer, thank you for your comment. We have addressed all the critical issues described in the first two comments. We hope to improve this new version of our manuscript, supporting a credible contribution.

Reviewer 2 Report

Comments and Suggestions for Authors

1 why L. casei Shirota was selected as a control?Please provide a proof.

2 why the result of the Bile salt and acid resistance has not been represent as figure? It would be more visual.

3 the abstract cant summarize the entire text well, please improve it.

Comments on the Quality of English Language

None

Author Response

Comment 1: why L. casei Shirota was selected as a control?Please provide a proof.

Response 1. Dear reviewer, Thank you for your concern. Nowadays, Lactobacillus casei Shirota is one of the most important commercialized probiotic bacteria worldwide. The Scopus database reports 68 cites, including the search words “Lactobacillus casei Shirota AND probiotics,” in the last five years. This probiotic bacterium is one of the most extensively used and highly studied.

In addition to the above relevance of this bacterium, we used it as probiotic control as previously, for the characterization of another LAB isolated from fermented pulque: DOI 10.1186/s40064‑016‑2370‑7

For this experiment, we isolated Lactobacillus casei Shirota directly from a commercial Yakut beverage by plating an aliquot in MRS (DIFCO) plates and incubating at 37 ºC for 24 hours. The resultant colonies were analyzed for purity and tested for catalase. The identity of the bacteria was identified by sequencing the complete 16S rDNA gene.

Comment two:  why the result of the Bile salt and acid resistance has not been represent as figure? It would be more visual.

Response two: As suggested, in the new version of the manuscript, we graphed the results of resistance to pH and bile salts. We agree that the graph gave a more visual appreciation of these results. New Figure 1.

Comment 3: the abstract can’t summarize the entire text well, please improve it.

Response 3: We modified the abstract section and considered that the new version summarizes properly the content of the entire manuscript. We hope to find an improved abstract in this new manuscript.